# A co-produced review of the experiences of Black male detention under mental health legislation: Challenging discrimination in psychiatry using The Silences Framework

Kim Heyes[1]*, Isaiah Brodrick[2], Debbie Best[1], Kenny Thompson[1], Caroline Leah[3], Eula Miller[1], Elaine Craig[1], Laura Serrant[1], Jeremy Dixon[4], Anna Bergqvist[5], Tella Lantta[6], Hári Sewell[7], Clair Tourish[1], Rachel Whyte[8], Joy Duxbury[9], Alina Haines-Delmont[1]

1 School of Nursing and Public Health, Manchester Metropolitan University, Manchester, United Kingdom, 2 Gaddum, Manchester, United Kingdom, 3 Department of Social Care and Social Work, Manchester Metropolitan University, United Kingdom, 4 Centre for Adult Social Care Research, Cardiff University, United Kingdom, 5 Department of Philosophy, Manchester Metropolitan University, Manchester, United Kingdom, 6 Department of Nursing Science, University of Turku, Turku, Finland, 7 HS Consultancy, Kington, Herefordshire, United Kingdom, 8 Department of Psychological Medicine, King's College London, London, United Kingdom, 9 Research for the Institute of Health, University of Cumbria, Lancaster, United Kingdom

* k.heyes@mmu.ac.uk

## Abstract

The number of detentions under mental health legislation is growing, disproportionately impacting Black men. Previous research into the over-detention of Black people is repetitive and solutions to reduce disparities are ineffective, not enacted, or outdated. This review is original and novel in using a lived experience lens within the Silences framework, to interpret and validate review findings and make actionable recommendations to enable change, reduce Black men's detention rates and improve experiences. The systematic review searched three databases: EBSCO, ProQuest, and PMC. Search terms included: ethnicity: Black African Caribbean; gender: male; and detention: detained under mental health legislation. Searches were conducted in September 2021 and February 2024 and included papers from 2000 to 2024. The review was conducted using the NIHR systematic review protocol. Searches resulted in 15,300 papers, which were reduced to 34 papers for inclusion in the thematic analysis. People with lived experience on the review team explored the Silences missing in the literature and co-developed the findings and recommendations. Three themes were developed and are presented as 'Screaming Silences' - a concept that amplifies what is known (by patients, family and friends, professionals, and others), but is not explicitly discussed within literature: (1) contextual identity; (2) culture, spirituality, and religion; and (3) power, language, and communication. People with lived experience discussed what these themes meant to them. Their views are key to flipping the narrative, and support change for Black men compulsorily detained in mental health settings. Findings show that academic understanding of the detrimental treatment and care of Black men has barely changed in twenty years. The recommendations centre

**Data availability statement:** The list of references includes all of the papers used in this study.

**Funding:** Funding was received from the National Institute of Health Research (NIHR), award number NIHR201715 awarded to AH-D, JD9, KH, CL, EM, LS, JD4, AB, EC, HS.This study/project is funded by the NIHR Policy Research Programme (NIHR201715). The views expressed are those of the author(s) and not necessarily those of the NIHR or the Department of Health and Social Care..

**Competing interests:** The authors have declared that no competing interests exist.

on: patient involvement and clear communication; reducing disparities through anti-discriminatory policies and practice; the promotion of cultural competence; community campaigns, collaboration, and support for carers; monitoring and auditing; and improving future research through co-production.

## Introduction

Black male experiences of mental health detention throughout the world are subject to bias, discrimination, and mistreatment [1]. This research presents arguments from thirty-four high-quality papers, incorporating some of the most important research in this field from over two decades. By introducing the Silences Framework to our systematic review methodology, we are presenting a novel and world-leading approach to tackling racism within psychiatric services using the voice of lived experience. The research reviewed in this paper highlights that changes to services in the Global North are not yet being made in a substantial enough way to prevent discrimination. Giving a voice to those who have not had one in this sphere is vital to understanding how to change services for the better. Although this research was conducted in England, we suggest solutions to policy and practice that can be incorporated into the new Mental Health Act (1983) [2], and world-wide.

Mental health services are in crisis. Data in England and Wales alone has shown a 20 per cent increase in patient detentions over recent years (2014–2016) [3], prompting reviews of the Mental Health Act (MHA) [2,4] in England and Wales [5,6], and the Government's Draft Mental Health Bill [7]. Of particular concern was the over-representation of Black African Caribbean people under the MHA, an ethnicity marker defined by the UK National Health Service (NHS) [8]. The experience of Black men was viewed as particularly concerning, with the perception of them as "big, Black and dangerous" being a major worry for people with lived experience [8,9]. Black African Caribbean men are more likely to encounter the mental health system via the criminal justice route than their White counterparts [6,10], this has led to calls for initiatives to reduce the use of section 136 which gives police the power to take people to a place of safety for a mental health assessment. The most recent figures show that 65.3 per 100,000 of Black people were detained under section 136 compared to 29.1 per 100,000 of White people [11,12].

However, data shows over double the number of Black people with mental ill health die in custody than any other demographic [13]. Studies have compared differences in the rates of detention across ethnicity by examining data showing that Black people are more likely to be detained for assessment or treatment under the MHA. Statistics in England from 2023-2024 show that 242.3 per 100,000 Black people detained under the MHA in this year, in comparison to 68.4 per 100,000 of White people [14]. At over three times the rate of detention, it is clear that racial disparity is not being challenged effectively in mental health services [15].

The need to understand what is happening during mental health detention for Black men requires creative methodologies, as previous research in this area is not being used to direct practice or policy. The Silences Framework [16] was developed for marginalised groups to be able to speak their truth about their experiences within healthcare. Through this framework we can challenge what is 'known' (by conducting a systematic review) by finding out what is actually happening to Black men (including people with lived experience as co-producers of the research) within a lived experience epistemological and methodological model that is on par with other clinical frameworks [17,18]. Most notably, a recent paper by Solanki et al. [19] conducted semi-structured interviews with twelve Black ethnic inpatients. Solanki et al's [19] findings mirror that of the Silences discussed in relation to our systematic review and showcase how

listening to the person with lived experience brings a depth of understanding that outweighs any longitudinal quantitative studies. This research demonstrates that qualitative data, which is sometimes considered less valuable, can be of higher quality when it includes the perspectives of those experiencing the issue. In this study, the Silences Framework gives a voice to Black men who have been detained under mental health legislation, and their parents and carers.

Despite the wealth of research into racial differences in involuntary detention rates, the same conclusions are being made by much of that research but the narrative of the racialised lived experience is yet to be seen. As has been recommended in many of these studies and perfectly articulated by The Sainsbury Centre for Mental Health (p 12) [20] "an important first step to improved service provision is engaging with service users". This paper does exactly that through co-production using the Silences Framework, and we urge practitioners and academics globally to do the same. While there are reviews capturing evidence with regards to experiences of detention under mental health legislation, to our knowledge this systematic review is the only one that includes people with lived experience as co-producers of the research.

The research question asked: what are the experiences of Black African Caribbean men who have been detained under mental health legislation? The objectives were twofold; initially to explore in what context does detention under mental health legislation occur for Black African Caribbean men; then to identify whose voices are missing from the evidence and what questions remain unanswered through the Silences Framework.

## Methods

Co-producing a systematic review including people with lived experience is a novel approach, and there is currently no framework that determines how this is conducted or written for publication. Therefore, we have written openly about the way our research was conducted and have included as much explanation as possible in the hope that moving forward, a framework for co-producing research may be designed. The authors epistemological stance for this study was a form of critical realism [21]. The systematic review was conducted with a realist subjective lens which interprets the findings via real-life experiences through the Silences Framework [16]. As there were experts by experience in the research team it was important to the authors that the main themes in the systematic review were explored through structured discussions focusing on the impact of the people who had been through these life experiences. All of the authors values were therefore considered in the reading of the data and space was given for each researcher to be reflexive as they read through the articles. Although the analysis was deductive, the Silences Framework allowed for an inductive approach which was experiential rather than based on the knowledge of the existing research. Critical realism takes into account systems and social structures which was important to the research team when analysing the findings from the systematic review. This required the team to use a thematic approach to the analysis rather than focusing on the quantitative data. The findings centre on the qualitative evidence, despite the majority of the included papers having a quantitative focus.

Studies were initially included if they were published from 1983 to 2022, published in English and including a specification of ethnicity (using Boolean terms) where Black, mixed heritage, African, or Caribbean is stated OR specifying the gender as male, OR specifying detention. This was to ensure all possible literature was captured, and we would limit this at the full paper stage, ensuring the papers had Black (or any associated words), male, and detention under mental health legislation, stated within their study. Exclusion criteria included papers not in English, and did not specify ethnicity, was about females only, did not include detention, and where participants were under 18 years old. The systematic review was registered with Prospero (CRD42022274045).

An iterative 5-step search strategy was adopted to identify studies:

1. an online search for systematic reviews,

2. grey literature search;

3. search appropriate journals;

4. reference lists from systematic reviews and meta-analyses checked for additional relevant studies;

5. Workshops enlisting advice from stakeholders including Black African Caribbean men who have experience of being detained, other professionals, lay people and experts as required.

Initial searches were decided upon in conjunction with librarians from the NHS and UK Police and the results were imported into Covidence [22]. S1 Appendix shows the search strings used. An updated search was conducted using the same databases and search strings in February 2024.

After duplicates were removed, a total of 15,300 papers were found which were uploaded into the management tool Covidence. Covidence was set-up so that each paper had to be reviewed by two people, with a third person reviewing any conflicts. Pragmatically and due to the time/ person power constraints of the research we needed to reduce the numbers of papers, so the initial search criteria were reduced to exclude research within criminal justice and papers published after 2000. The total reduced to 636 papers after the title and abstract screening and by removing papers using the additional exclusion criteria. A further search was conducted after a workshop with experts by experience in November 2021. These added the words spiritual, religion, psychophobia and a further 42 papers. After the full paper review, 72 papers matched the streamlined inclusion criteria. See Fig 1 below for a summary.

We used the Mixed Methods Appraisal Tool (MMAT) [23] to conduct a quality assessment at this point. Papers that scored low (bias) across all seven questions by two reviewers were included in the final study (a score of 7 indicated acceptance). A further search was conducted in February 2024 to capture any studies that had been published since 2022. This resulted in two more papers. The final 34 papers are summarised in Table 1.

Two reviewers conducted an equality appraisal based on a previous version by Bhui et al. [24], providing a score on how each included paper describes ethnicity, gender, and detention (S2 Appendix). This equality appraisal showcased the differences in the explanations of these demographics and therefore the difficulties involved in comparing the data. As this is not an approved measurement tool, we did not use it to exclude papers, but felt it was an important to include how vastly different demographics are recorded in healthcare, and how difficult that makes it for researchers to use the data to state anything with any certainty about equality, diversity, and inclusion issues. Systematic reviews were given a rating of N/A (not applicable) as the data was too varied.

The Silences Framework [16] was used as an overarching framework for the whole review. We used 4 stages of the Silences Framework to gain an understanding of what silences currently exist for Black men who have been detained under Mental Health legislation. Stage 1: Working in silences: a systematic literature review to provide a base from which we begin to draw out what is unknown or unsaid about race, ethnicity, and compulsory detention. Stage 2: Hearing silences: reflexive spaces dedicated to listening and hearing individual experiences. Stage 3: Voicing the silences: the analysis of the findings was continuous and cyclical. Initial findings were presented with attendees afforded the opportunity to give feedback. Stage 4: Working with 'silences': Researchers reflect on the potential impact of the findings, and what steps can be taken to achieve the goal or aim of the project.

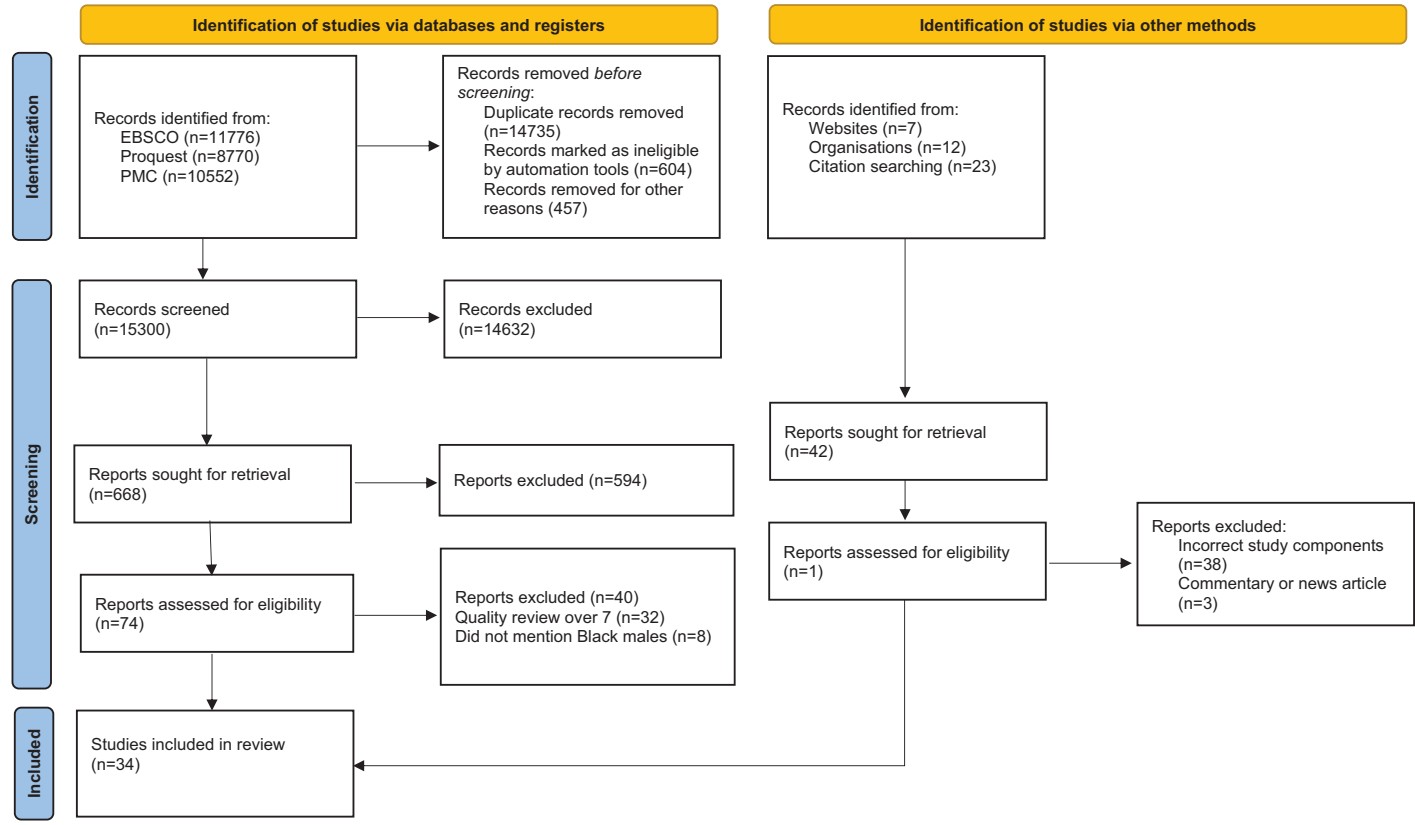

**Fig 1. PRISMA diagram depicting the stages of the screening process through to the final included number of papers.**

The four stages are cyclical, and each stage is to be considered several times as necessary throughout the life of the research. Through adopting a Silences Framework [16] and a truly authentic empowerment model of co-producing the research with people with lived experience, (Black people who have been detained under mental health legislation or caring for somebody who is, or has been detained), and mental health professional in these spaces, we collectively and continually questioned what was missing from the research. This authenticity was based on the racialised lived experience that was embedded this into the research from pre-production to co-production.

During this systematic review, three people with lived experience participated in conducting the review and co-developing the findings. Some of the references to their words are left intentionally ambiguous as we have an ethical responsibility to protect our colleagues and co-authors. The silences explored in this paper are the three most prolific themes arranged from a list of 50 subthemes. This was agreed in collaboration with all of the authors. Of the three people with lived experience on the team, one was a Black man with experience of mental health detention and two of those people have been carer to a Black man who had been detained. Other members of the research team also had lived experience, had worked as mental health professionals, or both. More people with lived experience contributing to this systematic review could have generated richer feedback from multiple perspectives, and possibly different silences.

We held a total of four workshops. The first workshop was in November 2021, to build a coding framework utilising the papers from the full paper search. Six main themes were

**Table 1. Table of included studies.**

| Lead Author and date | Country | Method | EDI Score (0=Excellent, 5=Unclear) |
|---|---|---|---|
| Alexandre 2010 | Portugal | Quantitative | 2 |
| Bansal 2022 | UK | Meta-ethnography | N/A |
| Barnes 2008 | USA | Quantitative | 5 |
| Barnett 2019 | Multiple | Systematic review and meta-analysis (Global) | N/A |
| Bhui 2003 | UK | Systematic review, narrative synthesis and meta-analysis | 5 |
| Bhui 2015 | Multiple | Systematic review (UK and USA) | N/A |
| Bolden 2005 | USA | Quantitative | 2 |
| Bookle 2011 | UK | Case-Control design | 1 |
| Boydell 2010 | UK | Quantitative | 2 |
| Chakraborty 2009 | UK | Qualitative | 0 |
| Coid 2000 | UK | Quantitative | 2 |
| Commander 2003 | UK | Quantitative | 0 |
| Evans 2017 | UK | Quantitative | 2 |
| Halvorsrud 2018 | UK | Systematic Review | N/A |
| Henderson 2015 | UK | Quantitative | 1 |
| Keating 2004 | UK | Qualitative | 0 |
| Kaselionyte 2019 | Multiple | Systematic literature review | N/A |
| McBride 2021 | UK | Quantitative | 0 |
| Mfoafo-M'Carthy 2014 | Canada | Qualitative | 3 |
| Mohan 2006 | UK | Quantitative | 1 |
| Oduola 2019 | UK | Quantitative | 2 |
| Oluwatayo 2004 | UK | Case note reviews | 2 |
| Raleigh 2007 | UK | Quantitative | 2 |
| Rotenberg 2017 | Canada | Quantitative | 1 |
| Saltus 2013 | UK | Quantitative | 3 |
| Singh 2007 | UK | Systematic Review | N/A |
| Singh 2014 | UK | Quantitative | 0 |
| Sohler 2004 | USA | Quantitative | 2 |
| Solanki 2023 | UK | Qualitative | 0 |
| Valenti 2014 | UK | Qualitative | 2 |
| Wanchek 2012 | USA | Quantitative | 3 |
| Watson 2015 | UK | Quantitative | 1 |
| Weich 2017 | UK | Quantitative | 3 |
| Whaley 2004 | USA | Quantitative | 2 |

created from this initial review: 'Variations in Pathways', 'Details Surrounding Detention', 'Misdiagnosis', 'Sub-demographics within Blackness', 'Poverty and Social Influences', and 'Legislation'. During the workshop, it was agreed that the themes portrayed a realistic view of what it was like for Black men who had been detained under mental health legislation, however, what was said did not go 'far enough'. This meant that although interesting points are raised within the literature, the people with lived experience felt that the researchers were too cautious with their interpretations, or did not include enough voices of the participants.

Research silences identified by lived experience co-researchers included religion and how empowering spirituality has been for some, the role of the police in detention, and the healing that takes place outside of services and often by oneself. This led to a second literature search to specifically find literature that included the search terms: 'psychophobia', 'spiritual

awakenings' 'alternative spirituality' 'mental health advocacy' and 'mistrust'. After going through the search process again, including quality review, one more paper was added to the final literature.

After collaboratively conducting a full review of the literature, a second and third workshop were held in April 2022 to produce the final themes of the paper. Analysis of the papers was through thematic analysis [25], and was conducted through hybrid (online and in-person) workshops. Two people read each paper and extracted the main findings. Flexibility of thematic analysis allows for the generation of themes across various methods of research and across multiple positionalities and theoretical epistemologies. The team read through a portion of the papers each before coming together to discuss the data extraction. This took place in the form of collating themes from each paper and determining which were the ones that came up the most. The second workshop involved the whole research team and produced 50 subthemes with four potential overarching themes and Silences found in the literature. The third workshop included two researchers and two people with lived experience from the review team. The people with lived experience lead on reducing the 4 overarching themes to three, alongside articulating the corresponding Silences. The initial themes and subthemes are presented in S3 Appendix.

The first draft of the review was created in September 2022 at a two-day writing retreat and included six members of the research team, two of which were people with lived experience. The writing retreat was created specifically to allow for undivided space and time to work on the first draft in a supportive, informal way as some of the researchers did not have as much experience as others in writing up a systematic literature review. This reduced any inequality felt within the team. During the retreat we also drew on the expertise of the Silences Framework method from its creator [16].

On day one we familiarised ourselves with the themes created from the literature review and discussed the initial silences. On day two Professor Laura Serrant, the creator of the Silences Framework, led discussions about finding the silences within the literature. Before the end of the second day, we had created the initial structure, themes, and silences for the systematic review paper. Versions of the paper were then created collaboratively through multiple online and in-person meetings, phone calls, and emails.

Researchers have a moral and professional responsibility to avoid harm to all study participants in any study. For experts by experience to feel comfortable about voicing silences, we adopted a reflexive approach where potential sources of sensitivity were understood, identified and the potential impact considered and proactively planned for as part of the study design. Distressing topics that emerged during the formulation of the systematic review led to people with lived experience needing time away from the project and mental health support from the qualified counsellor on the team. While a counsellor was available to all those involved in the project, more robust mental health support could have been developed in collaboration with people with lived experience prior to the start of the project.

## Results

Most of the papers (n=23) were based on studies conducted in the UK, mainly in England, however many of the studies included Wales and some included Scotland and Northern Ireland. Five studies were conducted in the USA, two in Canada and one in Portugal. Three systematic reviews used studies from multiple countries, with another two systematic reviews including papers from the UK only. There were 21 quantitative studies, five systematic reviews, five qualitative studies, two case study designs, and 1 meta-ethnography. Whilst the exploration of experiences predominately requires a qualitative answer, we have synthesised the quantitative and qualitative data to provide a more holistic overview. Quantitative data

was thematised by using the subheadings to the data commentary, much in the same way that qualitative data uses subheadings to describe themes.

A final writing workshop was held to address all of the themes identified from the systematic review, and identify the silences within them. The final themes were: Contextual identity; Culture, spirituality, and religion; and Power, language, and communication. These have been presented as a discussion of the systematic review findings, before voicing the silences relating to those points. This approach helps illuminate what is not being said in current research. The themes presented below is therefore structured beginning with information from the existing literature, as analysed by the full team, followed by a discussion of the silences that colleagues with lived experience felt was missing.

## Contextual identity

Within health services, the demographics used to describe patients are not individualised, but grouped into crude factors that only help to identify trends. This grouping together of data means that identities can be seen as monolithic, or as "one size fits all". This can be problematic in mental health services as there are many interrelated factors present in people's lives that could mean that misdiagnosis occurs, as a diagnosis could be based on the presenting factors alone. Black men in mental health services seem to have exactly this issue, and the included papers explore how their identity is described in relation to detention, specifically compulsory detention under mental health legislation across the Global North.

The included literature reports that Black men are more often detained through compulsory or involuntarily detention under mental health legislation than patients from other demographics [1,10,26–44], and the method of detention is likely to be through contact with the criminal justice system [1,10,27,30,42,43]. Black men who have been detained are usually younger than their White counterparts, are from socially deprived and minority-dense areas, are unemployed, have reached lower educational attainment, are single, live alone or in supported accommodation, and are more likely to be in insecure housing or are homeless [1,28,30,32,33,36,44–53].

Black men within these studies were also less likely to have a general practitioner (GP), have regular medical check-ups, and were more likely to mistrust healthcare professionals [1,10,27,29,34], despite having longer symptom durations before being detained [33,39,42]. Where a GP (or equivalent) was seen regularly, there were lower rates of compulsory detention [39,42,43]. This highlights that there are deep-seated systemic issues when it comes to accessing help and support in all areas of healthcare services for Black men.

Ethnicity was not found to be a statistically significant part of detention, or worse experiences of detention, in several studies [1,40,42,44,47,54]. Barnett et al., [1] posit that the studies in their systematic review found that there is a lack of primary evidence for much of the reasons given behind racial disparities. Singh et al. [46] believed that the higher detention rates could be due to the distribution of ethnic groups around the sites that their study was conducted within rather than other factors previously thought to be problematic, and Raleigh et al.'s [40] and Rotenberg et al.'s [42] studies both found no statistically significant evidence that there was an association with diagnosis or worse experiences of psychiatric care. Potential explanations, however, seem to coincide greatly with the identities of Black men as identified in the 34 included studies. Barnett et al. [1] for example states that increased rates of detention for all patients included factors such as the increased prevalence of psychosis, perceived risk of violence, increased police contact, absence of or mistrust of general practitioners, and ethnic disadvantages. Similarly, Singh et al. [47] describe factors related to compulsory detention including a diagnosis of a serious mental illness, presence of risk, living in supported accommodation, and living in London. These observations, whilst not being statistically significant

in these studies are commensurate with a multitude of research studies, many of which are included in this analysis.

Bansal et al. [55] terms having a mental illness as a 'form of social death' (p21). This highlights a further struggle for Black males being detained, as they feel as though they may be shunned by their community. Fear of being stigmatised in their community is a pressing issue for people with mental ill health. Solanki et al. [19] also described how some participants felt that their family and community stigmatized mental illness. The perceived lack of support outside of detention is problematic as the risk of being reintegrated into the community with an identity changed through a mental health diagnosis may be almost as fear inducing as being taken into mental health detention initially.

This review highlights the enduring disparities that exist within mental and general health services for Black men dealing with mental ill health. The research shows that identity is complex and multi-faceted. This means that although statistical significance may not support the difficulties that Black men face, what we need to understand is that the relationship between the contextual factors of compulsory detention and being a Black man is clearly shown [1,19,28,29–31,35,39,45–47,51–55]. Intersectional identities need to be understood to address the holistic issues experienced by Black men. It must be assumed that the individual may have experienced all these contextual factors, or none of them, with a wide spectrum of experiences in between [10,19,27,43]. Many of the included papers support the view that people should be treated as individuals and support should be tailored to their needs based on an understanding of culture, societal, and health needs [1,10,19,26,27,32,34,50,53,55].

## Identification of silences within contextual identity

A number of silences were identified around contextual identity, centering on the view that Black people are grouped as one and assumed to have the same or similar identity to one another. Currently, the literature reports on Black people's experiences only through the lens of their Blackness, a reductionist approach that fails to recognise the complexities of their self-identity; as such, biases against gender, sexuality, and class are not explored in tandem with race in much detail or beyond statistics. This does not adequately explore intracommunal differences among Black people. Some of the included studies documented the correlation between poverty and detention rates, and differences in detention rates among men and women, but no papers spoke to whether the negative experiences of Black men are exacerbated or multiplied by other identities such as being queer, or from a working-class background, for example. Seeking to understand the racialised experience in a vacuum can contribute to a cycle of silencing by not considering the narratives of those who have multiple marginalised identities. Person-centred care must seek to understand and care for the whole person beyond their race, allowing for a nuanced understanding of gender, sexuality, and class. In an example provided by an expert by experience, they explain that not only are they Black, but they are also a product of their upbringing, their sexuality, and much more. Those identities matter. They not only shape one's behaviour and perspective, but they will also shape the behaviour and perspective of those who provide mental health care, similar to the ways Blackness can be perceived in services.

## Culture, spirituality and religion

Only six papers [24,26,49–51,55] explicitly explore the role of culture, religion and spirituality in Black people's experiences of mental health services, however these topics featured heavily in the discussions of the authors. The authors believe these factors can be misinterpreted and

misunderstood which can lead to diagnostic errors and diminish the importance of spirituality within healing.

Clinical terminology was found to medicalize the view of spiritual practice by patients within psychiatric services [26,55]. The healthcare practitioners framed the experiences of patients as a part of their mental health diagnosis and posited this as evidence of poor insight and judgment. These 'features' were included in patients' notes and recorded as factual information. The notes omitted to record any personal or cultural information for context. An alternative discourse offered in Kaselionyte and Gumley [26] review found that first person accounts and interpretations of spiritual experiences were accepted and supported through listening, the use of spiritual guides or teachers and grounding techniques to promote self-healing.

Bhui et al.'s [24] study of therapeutic interventions to improve communications between Black and minoritised patients and professionals in psychiatric services found that effective interventions considered personal stories, cultural adaptions, and empowerment. Valenti et al. [49] found that patients appreciated it when their cultural norms and religious beliefs were respected during their hospitalisation.

Whaley's [50] study of paranoia in Black African-American patients found that there is a difference between the mistrust that the patient has culturally and interpersonally, and this is difficult to determine in the psychiatric setting. Interpersonal mistrust may be presented as fear of what White clinicians may represent to them in the wider social environment, which could be detrimental if the individual has experienced racism or prejudice. Cultural mistrust could be alleviated by having a clinician with a shared cultural perspective as the patient may feel more relaxed knowing that there is a shared understanding. Whaley [50] stated that it should be expected that Black patients raise complaints about White clinicians, but that these should be fully explored rather than dismissed as part of paranoia, delusion, or a sign of mental illness. The study found that there was no reason to associate cultural mistrust with a higher level of violence, as had been previously indicated by studies.

Services that do not utilise spiritual or religious practice as part of the healing process could be making issues worse for the patient and enhance the trauma of being detained [55]. Bansal et al.'s study found that by not considering people holistically and adhering to a social model of mental illness in addition to the medical model, mental health services are contributing to epistemic injustice, oppression, and discrimination [55]. This study also highlights that Black and minoritised mental health staff do not feel empowered to speak up or intervene on behalf of the patient when it comes to culture, spirituality and religion [55]. The authors state that the evidence from the meta-ethnography indicates the answer to the issue may not therefore lie in recruiting a more diverse workforce [55]. However, it could alternatively be interpreted as valuing the experiences of a diverse workforce is therefore vital, and staff should be encouraged to share their knowledge in a supportive environment. Instead of assuming that a more diverse workforce is not the answer, it may be that targeted recruitment in more leadership positions may enable Black and minoritised staff to feel more empowered.

All mental health professionals should receive training in cultural competence to better understand and address the unique needs and preferences of patients from diverse ethnic backgrounds [8,24]. This training should focus on improving communication and addressing potential cultural barriers. Improving mental health care for Black men involves culturally adapting interventions, prioritizing patient-centered communication, and conducting economic evaluations to support evidence-based policies and practices [24]. This includes utilising a wide range of knowledge systems within evidence-based medicine to accommodate various understandings of spirituality and acknowledging the co-existence of different perspectives by incorporating them into clinical practice and research [26]. Understanding of

mental health within family members and the community could also reduce stigma and fear [55]. Gaining input from community leaders, advocates, and individuals with lived experience, can help to tailor services to the unique needs of Black men and address cultural and language barriers [50].

## Identification of silences within culture, spirituality and religion

Voicing the silences when reflecting on matters of culture, spirituality, and religion, was very personal for the experienced members of the research team. On the whole, this topic was left out of academic and clinical studies. It was believed by the people with lived experience that the topic was normally deemed too sensitive and difficult to address within the academic literature, yet it was the most important issue for them.

Those with lived experience on the project explained that spiritual practices and beliefs that exist outside of Judeo-Christian religions are often misunderstood and pathologized in mental health services. Behaviour or beliefs that do not align with Western concepts of spirituality become a part of the individual's mental health diagnosis or evidence of a mental illness, which strips them of their autonomy by attributing a choice and a belief that could ground them to their mental state. One example that was shared included the pathologisation of a detained person who had been isolated because they spoke to themselves in what was described as a 'strange' way at night-time. This was seen as a symptom of the mental illness, but the ward staff were confused by this presentation. After several days, a senior mental health nurse saw this for the first time and recognised it as glossolalia - or 'speaking in tongues' - being used as a method of prayer. The lack of knowledge by other members of the nursing and ward staff meant that the person was detained for longer than necessary and there was a risk of them being diagnosed and medicated wrongly.

Similar to findings reported by Kaselionyte and Gumley [26], we acknowledge that some behaviours or beliefs can be harmful. However, the belief that there is only one truth that is essentially decided by the practitioner, continues a cycle of silencing that pathologises beliefs that could be an essential part of the individual's recovery or a barrier against institutional racism.

Another overarching silence that resonated among the authors was the idea of professional curiosity; the desire to remain open and curious about who the patient is as a human being, their life history and how that contributes to their current behaviour and mental state. It is essential that those responsible for formal diagnosis have a willingness to understand traditions and cultural differences. It is equally important that all professionals are trauma informed and work from a place of curiosity, encouraging practise that aids recovery and helps ensure that individuals' thoughts or emotions are not medicalised and or inappropriately attributed to mental health.

The perception that Black people all share the same views and experiences of mental health services contributes to the misunderstanding of their experiences. This contributes to a cycle of silencing experiences that do not align with more accepted or well-known experiences and leads to reductionist models that silence and disempower the racialised lived experience in research, mental health policy and practice. Though there is a collective solidarity among Black communities in the UK, there is also intra-communal segregation between African and Caribbean communities, even some division within African and Caribbean communities based on country, island, and religion. The findings gathered from the literature and the silences identify the importance of culturally appropriate care that is malleable, open to change, and does not assume the cultural needs of a patient just based on their race or background.

## Power, language and communication

Fear is often at the centre of the distrust experienced by Black patients, this could be fear based on experiences of racism, fear of the stigma attached to being diagnosed with a mental illness, and the fear one might experience when entering mental health services [51]. Most notably, participants in one study attributed some of their fears of mental health services to the ways services mirror other oppressive institutions such as education systems, the police, and the criminal justice system [52]. The oppression felt by participants in their everyday lives was believed to be replicated in mental health services, with many participants expressing fatigue in fighting against racism. In Whaley's [50] study scoring high on a cultural mistrust scale was positively correlated with a mistrust of mental health services even if they also believed that White doctors were better trained than doctors of colour, highlighting the importance of patients' trust in staff and services. Racial disparities in detention and diagnoses are evident throughout the included literature reporting that Black men are more likely to be compulsorily detained under mental health legislation [27,30,37,38,40,44,47], they are more likely to be admitted by the police or criminal justice system, they are most likely to be diagnosed with schizophrenia, schizo-affective disorders or psychosis [1,28,32,33,36,39], they spend longer in detention on average [34,41], and are more frequently re-admitted [1,39].

The included literature reveals serious racial disparities in psychiatric care around the Global North. Bias, whether conscious or unconscious, is affecting the lives of Black men and means that psychiatric services are not considering either cultural or racial factors when interpreting symptoms or implementing care plans [33]. As a result, there is a mistrust of the healthcare system, which often leads to delayed presentations at emergency or crisis settings instead of getting the care they need at the right time [29,46,51,53]. Many of the studies reported people feeling as though they were being coerced or treated differently due to their race, and this made it difficult to feel safe and trust the care they were being given [52,54]. Psychiatrists also stated in one study that they found it difficult to talk about race and culture highlighting the need for training [51]. Boydell et al.'s [48] study found that although patients did not agree with their medication and other aspects of care, their satisfaction levels with psychiatric service were the same as White people. Similarly, Raleigh's study did not support the view that Black people were treated worse in psychiatric care, and two systematic reviews stated that although racial stereotyping, alienation, mistrust of services, greater stigma, language barriers and poorer detection of mental illness were cited as factors relating to worse treatment, there was often no real evidence supporting this [40].

Communication from healthcare staff to patients was found to be coercive and oppressive, often using inaccessible language (medicalised 'jargon') which was perceived as power play. Black participants in a study by Chakraborty et al. [52] question their diagnosis, with one participant stating they have depression and not the schizophrenia that they have been diagnosed with, and another explaining their mental illness is a manifestation of physical pain. Providing some positive experiences for community treatment order (CTO) case managers, participants in a study by Mfoafo-M'Carthy [54] reported that having a good rapport with their case manager or having a worker who was understanding and supportive helped them feel less like a patient and allowed them to communicate freely. However, most of the patients in this study reported being treated negatively as a result of agreeing to be put on a CTO. They state that they had been coerced into agreement whilst not fully understanding the circumstances [54]. Many patients felt as though they lacked control over their treatment decisions, although this was dependent on the level of respect they got from the staff in the hospital [49], and Solanki et al.'s [19] study found that the lack of choices patients were given, resulted in them feeling a lack of support in the way that they wanted it. In Valenti et al.'s [49] study, patients reported abuse and behaviours from staff that negatively impacted on their experience.

### Identification of silences within power, language and communication

People with lived experience noted two silences regarding language and communication in services; the quality of language (whether language was appropriate or used to confuse and control), and communication used by staff and services, and the lack of curiosity from staff to understand the language and ways communication used by patients. It was also recognised that qualitative, in-depth studies reporting the perspective of both people with lived experience and professionals is scarce within the literature.

The language and ways of communicating used by services and professionals can be silencing for carers and patients and create confusion and conflict. People with lived experience reflecting on their experiences with services, recalled gaslighting language, staff and doctors that making them feel inferior and their concerns being ignored. One's environment and the way they are treated can tie heavily to their identity, the oppressive language used in services can chip away at someone's identity and what makes them who they are.

Language and behaviour are open to interpretation and there are often deeper meanings within words, language and communication. Without curiosity and trauma-informed systemic practice the deeper meaning behind the behaviour of patients can be lost, limiting the chance of a true rapport, or understanding between patient and worker, and hindering healing for the patient. People in services choosing to keep their head down or following rules they do not agree with can create a revolving door, where people leave services without their mental health needs actually being addressed, and equally, reacting aggressively out of fear or frustration will prolong their stay in services. When it feels like those are the only options available to you, the healing and recovery of the person cannot happen. Services would better treat people and reduce readmission rates if they fostered an environment where people felt safe and comfortable to express how they feel, both the good and bad. Staff members could be more engaged, empathetic, and curious, dedicated to empowering the patient, helping to address the hierarchical power balance between them.

## Discussion

Experiences of Black men who are detained under mental health legislation have not changed in meaningful ways for at least the past twenty years. This systematic literature review shows how research papers report the same issues repeatedly: Black men are younger, usually diagnosed with psychosis, have longer hospital stays, are more frequently readmitted, are unhappy with their treatment, and are more often detained through the criminal justice system than any other demographic. Their voices are missing from literature until now, and our unique Silences Framework systematic review has allowed for people with lived experience to voice their unheard experiences. The statistics show that the over-representation of Black men in mental health detention comes at a human and financial cost, demanding attention [20]. Our findings add further meaning to this. Presenting the findings from the existing literature alongside the voices of those with lived experience is a way of changing the narrative and exposing the silences [15–17].

Black men are presented within the literature as one collective community with similar beliefs, values, and social capital [1,28,38,39,42,45]. The silences that were discussed by people with lived experience showed that being 'treated as a monolith' meant that individual identities were ignored. By ignoring the intersectionalities that exist for Black men, there is a chance that misdiagnosis and mistreatment may occur, which is where racism, difficulties with staff and understanding the treatment given; and a lack of choice for treatment and support continues to be perpetuated [19,33,49,53,55].

The primary concern raised by many Black men in the literature and from people with lived experience, is about their specific experiences of *racial discrimination*. However, there is a need to also emphasise the importance of intersectionality while acknowledging that race-specific issues remain a crucial part of the narrative [56]. It is critical to clarify that Black men are most likely to be detained and suffer biases due to their Blackness; highlighting race as a primary factor and that these inequalities exist regardless of the intersectional identities.

This could be misinterpreted as a philosophical dilemma within the context of Black men's multiple intersectionality's. Blackness is a primary inequality driver but, it should not be so reductionistic that it is seen in isolation from other intersecting identities such as gender, sexuality, class, and disability. Intersectionality theory highlights the double disadvantage and inequality that is experienced by people with mental illness who belong to multiple stigmatized social groups [57–59]. Interventions aiming to reduce these inequalities need to be flexible and targeted towards intersectionalities (gender, sexuality, etc) rather than universal (Black men) to effectively address systemic inequality. Therefore, the desire to be seen as more than monolithic is not to take away the Black identity from the man but from a desire to be seen as more than a negative stereotype and re-address the stigma, and trauma rooted within this. Addressing these negative perceptions and racial biases is an important step toward reducing predetermined projections of mental health behaviours on to Black men during detention. Creating environments where people feel as though they are safe, they have a voice, and they are listened to will ensure that there is a reduction in fear, frustration and readmission [50,54–56].

The studies included in the systematic review were subject to a quality review to minimise the number due to the time afforded through the funding of this part of the wider project. This in itself is a form of silencing. During the quality appraisal all grey literature was excluded.

This review is unique in that it moves away from the standard approach to reporting systematic reviews. Instead, it allows for the insights of people with lived experience and highlights the importance of collaboration and being flexible and responsive to people's needs and 'meeting people' where they are at. We are eternally grateful to those who have been brave enough to share their personal stories, and their vulnerability and honestly throughout the process. The team have reflected upon their own positionality which has impacted both positively and negatively on the journey at times. We understand how important it is to learn from such experiences and we anticipate we will reflect more fully on these in future research papers.

There is undoubtedly significant scope for the mental health detention experiences of Black men to be substantially improved upon given issues highlighted in this review and concerns raised more generally over recent decades. We make six recommendations for policy and practice based on this research. These are: patient involvement and clear communication; reducing disparities through anti-discriminatory policies and practice; the promotion of cultural competence; community campaigns, collaboration, and support for carers; monitoring and auditing; and improving future research through promoting coproduction.

## Patient involvement and clear communication

Our research found that many patients felt that they lacked control in the decision-making regarding their treatment. Communication and assessment strategies that prioritise active listening, empathy, and understanding of patient's cultural and spiritual backgrounds, would promote a culture of respect and empathy within health and social care environments. These strategies have the potential to illuminate the nature of personal experience as well as the

social structures which cause or amplify mental distress [60]. Professionals should discuss the diagnosis with the patient, explore any reservations and their reasoning. For example, the Joint Committee on the Draft Mental Health Bill [61] has argued that reforms to the Mental Health Act in England and Wales should enable individuals to write an advanced choice document, stating their treatment preferences. This should include views on alternative treatments, therapies, social support and medicines as a valid form of care and treatment choices, or as part of a combination. Current evidence suggests that this type of advanced care planning has the potential to benefit black groups particularly and so these measures should be implemented within the new Mental Health Bill in England and Wales and should be adapted to other juridstrictions [62,63].

## Reducing disparities through anti-discriminatory policies and practice

The implementation and enforcement of antidiscrimination policies within mental health care settings would help to ensure that Black men receive equitable treatment, and that biases in referrals and admissions are minimised. These policies should address both direct and indirect forms of discrimination within the system. Policies should be transparent and regularly reviewed. Targeted interventions around compulsory admissions need to address factors such as cultural barriers relating to systemic racism, a lack of community support and understanding, and a lack of GP or healthcare access. Training aimed at raising staff awareness of anti-black racism, has the potential to acknowledge and validate black men's experiences, thus improving therapeutic engagement [64] and is supported as a mechanism to remedy racism within services by black service providers and community members [65]. Fear is a central reason that most Black men do not access timely interventions and why they are perceived as violent when being detained [51]. Having policies that actively reduce this fear by ensuring humane treatment of individuals in crisis, alongside community interventions and stigma reducing campaigns would be of benefit.

## The promotion of cultural competence

Cultural competency and understanding of spirituality and religion were lacking in the majority of the papers reviewed in this study, despite other research findings which indicate that cultural competence by mental health services is viewed as important by black people [66]. Bridging the gap between medical knowledge and spiritual healing can be affected through collaboration, integrating diverse therapeutic approaches, and involving those with lived experience to understand how mental health care can provide a more holistic experience to meet the needs of people from all backgrounds. Initially, this can be through integrating ongoing cultural competency training to healthcare professionals, collaborating with spiritual and cultural leaders [67], adapting existing interventions, and simply asking patients what they feel is the best treatment for their own mental health care [68]. Efforts should be made to create culturally appropriate care that considers all aspects of a patient's identity, this could include their race, nationality, gender, sexuality, or class for example. Care that only acknowledges one aspect of who they are may improve their experiences in some capacity, but still fails to provide true person-centred care [69].

## Community campaigns, collaboration, and support for carers

In several of the papers reviewed, community interventions including regularly accessing a GP, reduced the need for crisis interventions, leading to compulsory detention through mental health legislation. Through campaigns to reduce the stigma of mental health in communities that we know have a high level of Black male detention rates, particularly within the Black

African and Caribbean diasporas, people may start to seek support earlier. For example, storytelling interventions, which aim to amplify the experiences of clack people experiencing mental illness have been found to lower mental health stigma across demographic cohorts [70]. In addition to this, there needs to be an improvement to early intervention strategies. Utilising community and well-being champions [71] and working with faith-based has the potential to improve access to services [68]. Access to GPs, community-based mental health interventions, and support networks for at-risk individuals, should also be improved. Additionally, support for caregivers is essential, as they often have negative views of mental health services and experience difficulties in seeking help [72]. Such experiences are not universal and building effective relationships with carers can help them to support and advocate for the needs of their loved ones.

## Monitoring and auditing

Being aware of the issues around the disparities for Black men in mental health services should be subject to accountability. This means that monitoring and auditing should take place on a continual basis to ensure that psychiatric and mental health services are complying with changes to policy and practice and are able to respond to any other issues that may occur in a timely manner [73].

## Improving future research through embracing co-production

There should be investment in future research to understand and mitigate further disparities, with an emphasis on evidence-based policy development incorporating co-production in collaboration with experts by experience. The findings from the research, and the silences within research uncovered in this paper was led by people with lived experience of detention or of caring for someone who has been detained. Future research should always be developed in co-production with service users and carers who have lived experience to create meaningful and transformative change including at operational and strategic levels. Research conducted with people with lived experience has the potential to capture the mental health detention process from the perspective of the population that is actually impacted by services and gives Black people agency within academia and health and social care. However, whilst the value of co-produced research has been realised, it remains at early-stage and so further commitment is needed from both researchers and health and social care funders [74].

In conclusion, raising the voices of those that feel silenced is vital to changing services in a meaningful and sustainable way. Intrinsically racist practices are difficult to see as they are so embedded into the fabric of mental health services and are commonly dismissed as 'that is the way it has always been done'. By conducting this review in co-production, and embedding the experienced voice throughout, we have shown that although academic research is addressing disparities, it can continue to perpetuate damaging assumptions. By presenting a systematic review alongside a discussion of silences, this study goes some way towards helping practitioners, policymakers, academics, family and friends, and other experienced mental health service users, see how easy it is to invoke discussion and to begin to unravel harmful and racist practices.

## Limitations

This is the first time that we know of that anyone has tried to understand the silences that come from a systematic review. In some instances, this may not be determined as generalisable as the authors are a small group of experts and people with lived experience commentating on issues that are not largely reported on within so called high quality academic literature.

Such evidence may be viewed as less robust in traditional research terms. Nonetheless, these recommendations still have value, as they come from people with lived experience. Lastly, there was a notable imbalance in the location of studies, with most studies taking place in the global north. This situation reflects an imbalance in mental health research between the global north and south more generally [74], but means that our findings may not be easily applied to countries outside of this jurisdiction. Whilst recognising the disparities in mental health research funding internationally, our review identifies a need for greater research into the experiences of Black men in other countries given that systems for mental health detention vary across countries and that experiences of detention may also differ. Despite these limitations, we believe that our study has value as the first paper to utilise the Silences framework within a systematic review which will provide the platform for further work of this nature going forward.

## Supporting information

**S1 Appendix.** **Search strings from the initial searches.**
(DOCX)

**S2 Appendix.** **Ethnicity, gender and detention scoring table.**
(DOCX)

**S3 Appendix.** **Table of initial themes and sub-themes.**
(DOCX)

## Acknowledgements

The material in this paper draws on past research that has very much been at the forefront of research within the area of mental health, and in particular that of minoritised groups. We wish to thank the authors of these papers for their insight. We would also like to thank the people with lived experience that have contributed to our wider research.

## Author contributions

**Data curation:** Kim Heyes, Isaiah Brodrick, Caroline Leah, Jeremy Dixon, Anna Bergqvist, Elaine Craig, Rachel Whyte, Joy Duxbury, Alina Haines-Delmont.

**Formal analysis:** Kim Heyes, Isaiah Brodrick, Debbie Best, Kenny Thompson, Eula Miller, Laura Serrant, Jeremy Dixon, Anna Bergqvist, Joy Duxbury, Alina Haines-Delmont.

**Funding acquisition:** Kim Heyes, Caroline Leah, Eula Miller, Laura Serrant, Jeremy Dixon, Anna Bergqvist, Elaine Craig, Hári Sewell, Joy Duxbury, Alina Haines-Delmont.

**Writing – original draft:** Kim Heyes, Isaiah Brodrick, Debbie Best, Kenny Thompson, Caroline Leah, Eula Miller, Laura Serrant, Jeremy Dixon, Anna Bergqvist, Hári Sewell, Clair Tourish, Joy Duxbury, Alina Haines-Delmont.

**Writing – review & editing:** Kim Heyes, Isaiah Brodrick, Debbie Best, Kenny Thompson, Caroline Leah, Eula Miller, Jeremy Dixon, Elaine Craig, Tella Lantta, Joy Duxbury, Alina Haines-Delmont.

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
