## [Decision Letter · Decision Letter 0]

3 Sep 2024

PMEN-D-24-00122

A Co-Produced Review of the Experiences of Black male detention under mental health legislation: Challenging discrimination in Psychiatry using the Silences Framework.

PLOS Mental Health

Dear Dr. Heyes,

Thank you for submitting your manuscript to PLOS Mental Health and thank you so much for your understanding and patience as we have tried to secure reviews over such a long period. After careful consideration and looking at the reviewer feedback, we feel that it has merit but does not fully meet PLOS Mental Health’s publication criteria as it currently stands. Therefore, we invite you to submit a revised version of the manuscript that addresses the points raised during the review process.

Please address all of the comments raised by the reviewers that you will be able to see at the end of this email.

We look forward to receiving your revised manuscript and thanks again for your patience.

Kind regards,

Karli Montague-Cardoso

Executive Editor

PLOS Mental Health

Additional Editor Comments (if provided):

Reviewers' comments:

Reviewer's Responses to Questions

**Comments to the Author**

1. Does this manuscript meet PLOS Mental Health’s publication criteria ? Is the manuscript technically sound, and do the data support the conclusions? The manuscript must describe methodologically and ethically rigorous research with conclusions that are appropriately drawn based on the data presented.

Reviewer #1: Partly

Reviewer #2: Yes

2. Has the statistical analysis been performed appropriately and rigorously?

Reviewer #1: I don't know

Reviewer #2: Yes

3. Have the authors made all data underlying the findings in their manuscript fully available (please refer to the Data Availability Statement at the start of the manuscript PDF file)?

Reviewer #1: Yes

Reviewer #2: Yes

4. Is the manuscript presented in an intelligible fashion and written in standard English?

Reviewer #1: Yes

Reviewer #2: Yes

5. Review Comments to the Author

Reviewer #1: After a thorough review of the manuscript, I commend the authors for their innovative and comprehensive approach to addressing the experiences of Black African Caribbean men detained under mental health legislation. The manuscript is well-structured and addresses a critical area within mental health research.

However, to enhance the clarity, depth, and impact of the manuscript, I recommend a major review. This recommendation is based on the need for significant clarifications and enhancements in several key areas, including:

Introduction

1. In the abstract, you mention 34 papers, and in the second line of your intro you state 32 papers. I may be confused that these are high quality papers and the other two were not?

2. The first paragraph of the introduction sounds like a repeat of the abstract. While summarising is useful, consider restructuring to avoid repetition. You might start with a global perspective, narrow down to Black African Caribbean individuals, and then focus on Black males specifically.

3. The phrase “Of particular concern was the over-representation of Black African Caribbean people under the MHA” could benefit from a reference

4. Ensure consistent capitalisation of terms like "Black" and "white" for clarity and consistency for example you state: “Black African Caribbean men are more likely to encounter the mental health system via the criminal justice route than their white counterparts”. Same as the proceeding sentence: “The most recent figures show that 65.3 per 100,000 of Black people were detained under section 136 compared to 29.1 per 100,000 in those of white ethnicity” and this sentence “Black people detained under the MHA in this year, in comparison to the reducing 74.7 per 100,000 white people”

5. When you introduce the Silences Framework, it feels like part Methods section and part Discussion section. I would suggest most of the discussion elements be left to the discussion section.

6. NHS - in a UK context people may know this, however, in a global context may be useful when using it first time to use the full “National Health Service”?

Methods

1. May you give an explanation as to how you came to a thematic approach than focusing on the quantitative data.

2. In the second paragraph of the Methods section, is the use of 'OR' here meant to indicate criteria rather than explicit Boolean operators? Using uppercase 'OR' might make it seem as though Boolean operators are being explicitly applied in a search strategy, which could be confusing if that is not the intention.

3. May you provide a reference and description at first mention of “Covidence”?

4. There are some grammatical issues, such as ensuring subject-verb agreement. For instance, “the initial search criteria was” should be “the initial search criteria were.” Reviewing for grammatical errors and shortening some sentences could improve readability.

5. Just to comment I enjoyed reading this bit. Well done: Additionally, two reviewers conducted an equality appraisal based on a previous version by Bhui et al., providing a score on how each included paper describes ethnicity, gender, and detention. This equality appraisal is included to showcase the differences in the explanations of these demographics and therefore the difficulties involved in comparing the data. As this is not an approved measurement tool, we did not use it to exclude any papers, but felt it was an important to include how vastly different demographics are recorded in healthcare, and how difficult that makes it for researchers to use the data to state anything with any certainty about equality, diversity, and inclusion issues.

Results

1. Clarification is needed on the number and purpose of workshops mentioned. Distinguishing between different workshops or summarising the results more concisely could reduce redundancy

2. The results section focuses heavily on quantitative data despite your qualitative approach in the methods. Aligning these sections more closely could enhance coherence.

3. Stigma plays a significant role in mental health experiences. However, in your manuscript, it receives an almost footnote explanation without sufficient wider context. You briefly mention stigma and then quickly move on to state that identity is complex and multifaceted. This transition feels abrupt, and it’s challenging to understand the full impact of stigma on the complex identities of Black African Caribbean men in this context. A more detailed examination of how stigma intersects with and exacerbates the challenges of these multifaceted identities would strengthen your analysis. For instance, how does stigma related to mental health interact with other forms of stigma (e.g., racial or socio-economic) faced by Black men? Providing a more integrated discussion could offer deeper insights into their experiences

4. You mention, “Intersectional identities need to be understood to address the holistic issues experienced by Black men, and it must be assumed that the individual may have experienced all these contextual factors, or none of them, with a wide spectrum of experiences in between.” This is followed by, “This supports the view that people should be treated as individuals and support should be tailored to their needs based on an understanding of culture, societal, and health needs.” Are these two statements related to specific literature or results from your review? If so, it would be beneficial to explicitly link these statements to your findings or to existing studies. Currently, it feels like the first statement is a general comment about intersectionality, while the second is a related but separate point about individualised treatment. Clarifying how these points are supported by your results or the literature would strengthen your argument. For example, you could reference specific studies that highlight the importance of considering intersectional identities in mental health treatment and how this informs the need for tailored support.

5. You state, “Currently, the literature reports on Black people’s experiences only through the lens of their Blackness, a reductionist approach that fails to recognise the complexities of their self-identity; as such, biases against gender, sexuality, and class are not explored in tandem with race in much detail or beyond statistics.” However, isn't the stated aim of your review to address these very complexities and intersections? You begin by stating that Black people/men are most likely to be detained and suffer biases due to their Blackness. This seems to highlight race as a primary factor. Within the context of your intersecting concepts, it appears that you may be arguing against the reductionist view while simultaneously emphasising the significance of Blackness. This could be seen as a philosophical dilemma, but clarifying your stance could strengthen your argument. Are you suggesting that while Blackness is a significant factor, it should not be considered in isolation from other intersecting identities such as gender, sexuality, and class? Explicitly addressing this potential ambiguity and providing balanced explanations or examples from your findings could help resolve this apparent contradiction.

6. You state, “Seeking to understand the racialised experience in a vacuum can contribute to a cycle of silencing by not considering the narratives of those who have multiple marginalised identities.” While I agree with this statement, isn't there a risk of ignoring the significant inequalities faced by Black people, as highlighted in the earlier literature you included? These inequalities exist regardless of the intersectional identities. The primary concern raised by many Black individuals is that their specific experiences of racial discrimination are often overlooked in favour of other characteristics, which, while important, might dilute the focus on race-specific issues. This could make your study appear similar to the studies you were initially critical of, which tended to overlook the Black experience. It would be beneficial to balance your discussion by emphasising the importance of intersectionality while also acknowledging that race-specific issues remain a crucial part of the narrative. Providing clear examples and explanations of how both perspectives are integrated into your findings could enhance the depth and originality of your study.

7. The example you give, “In an example provided by an expert by experience, they explain that not only are they Black, but they are also a product of where they have been brought up, their sexuality, and much more. Those identities matter and will not only shape one’s own behaviour and perspective but will also most likely shape the behaviour and perspective of those who provide mental health care, similar to the ways Blackness can be perceived in services,” raises an important point. However, it could be interpreted as suggesting we should ignore their Blackness and focus on everything else. This is problematic because, as you acknowledge, the systemic issues faced by Black individuals often mean that their race is a primary factor in how they are treated, regardless of other intersecting identities. This example might be more appropriate in the discussion section where you could expand on this insight. You could explore how both the individual's broader identity and their race intersect to impact their experiences in the mental health system. This would allow you to provide a more detailed discussion on how systems need to consider both race and other intersecting identities to provide holistic and effective care. This is related to the two points above. I am not trying to silence their voice but it may be better amplified if it supports the narrative that you are trying to un-silence the voices of the Black experience.

8. I think your results section would benefit from the inclusion of quotes from the studies. Direct quotes can provide powerful, illustrative examples that bring the data to life and add depth to your analysis. They can help to highlight key findings and give voice to the participants’ experiences, making the results more engaging and impactful. Including quotes can also enhance the credibility of your findings by providing concrete evidence from the literature.

9. I suggest reviewing and clearly specifying the type of thematic analysis you are utilizing. Based on my understanding of your results, it seems you may have attempted to use reflexive thematic analysis. However, the results section contains a lot more discussion than actual analysis. It would be beneficial to ensure that the results section focuses on presenting the data and findings derived from the thematic analysis. More detailed analysis of the themes, supported by quotes and examples from the studies, would strengthen this section. Clarifying the methodological approach and ensuring a clear distinction between results and discussion will enhance the rigor and clarity of your manuscript.

10. You state, “The perception that Black people all share the same views and experiences of mental health services contributes to the misunderstanding of their experiences. This contributes to a cycle of silencing experiences that do not align with more accepted or well-known experiences and leads to reductionist models that silence and disempower the racialised lived experience in research, mental health policy, and practice. Though there is a collective solidarity among Black communities in the UK, there is also intra-communal segregation between African and Caribbean communities, even some division within African and Caribbean communities based on country, island, and religion.” These claims and statements require references to support them or at least a disclaimer indicating that these are your views. While it is understood that there are differences and similarities in experiences, my understanding is that your study has primarily focused on comparisons. You have emphasized that "not all Black people are the same," but in doing so, you may have overlooked the fact that Black individuals are more likely to suffer from the inequalities you mention. It is important to balance your discussion by acknowledging these pervasive inequalities while exploring intra-communal differences. Providing references or explicitly stating that these are your interpretations will enhance the credibility and clarity of your argument

11. You state, “Racial disparities in detention and diagnoses are evident throughout the included literature reporting that Black men are more likely to be compulsorily detained under mental health legislation24,27,34,35,37,41,43, they are more likely to be admitted by the police or criminal justice system, they are most likely to be diagnosed with schizophrenia, schizo-affective disorders or psychosis1,29,33,25,30,36, they spend longer in detention on average 31,38, and are more frequently re-admitted 1,36.” This is a very powerful statement, and the frequent references underscore its importance. However, I felt somewhat underwhelmed by the subsequent discussion, which seems to downplay the significant role that race plays in admission, diagnosis, and interactions with the criminal justice system. Given the strong evidence you have cited, it would be beneficial to highlight and delve deeper into these racial disparities

12. Your statement, “Bias, whether conscious or unconscious, is affecting the lives of Black men and means that psychiatric services are not considering cultural factors when interpreting symptoms or implementing care plans,” needs further clarification. Are you indicating that psychiatric services are not considering cultural factors, racial factors, or both? The sentence as currently written seems to conflate culture and race. It is important to clearly distinguish between these two concepts to avoid confusion. If you are referring to both cultural and racial factors, consider revising the sentence to explicitly state this. For example, “Bias, whether conscious or unconscious, is affecting the lives of Black men and means that psychiatric services are not considering either cultural or racial factors when interpreting symptoms or implementing care plans.

Discussion

1. You state, “Their voices are missing from literature until now, but the statistics show that the over-representation of Black men in mental health detention comes at a human and financial cost and demands attention.” Could you please clarify where these voices are explicitly included in the review? It would be beneficial to specifically indicate whose voice or view is being represented. I recall only one instance where a Black individual expressed wanting to be viewed beyond their Blackness, including their sexuality and upbringing. In the identified literature, it is not clear if these perspectives are from clinicians, researchers, or Black men themselves. Providing clear attribution of these views would enhance the clarity and impact of your discussion. For example, explicitly stating whether the perspectives are those of the Black men affected, the clinicians treating them, or the researchers studying them would provide a clearer understanding of the different viewpoints being presented

2. You state, “By ignoring the intersectionaities (CHECK SPELLING) that exist for Black men, there is a chance that misdiagnosis and mistreatment may occur, which is where racism, difficulties with staff and understanding the treatment given, and a lack of choice for treatment and support continues to be perpetuated.” I agree with this statement. Ignoring the existing racism may indeed lead to difficulties in staff providing appropriate care, support, and understanding of the treatment given. It would be beneficial to integrate this point with your earlier statements in the results section to provide a cohesive argument. I appreciate your subsequent statement, “Addressing perceptions and racial bias is an important step toward reducing predetermined projections of mental health behaviours on to Black men during detention.” Thank you for acknowledging the presence of racial bias. Ensuring that both intersectionalities and the direct impact of racism are addressed will strengthen your argument and provide a more comprehensive understanding of the issues at hand

3. You state, “The studies included in the systematic review were subject to a quality review to minimize the number due to the time afforded through the funding of this part of the wider project. This in itself is a form of silencing, and we did not include any grey literature.” I appreciate your honesty about this limitation. However, I did not get a clear reason as to why grey literature was explicitly excluded. Providing a rationale for this exclusion would help readers understand the scope and limitations of your review more fully. For example, you could explain whether it was due to concerns about the quality and reliability of grey literature, resource constraints, or other factors. This additional context would enhance the transparency and comprehensiveness of your methodology

1. One of your recommendations is “support for carers.” However, this aspect was not addressed during your review, either through the identified studies or through the voice of a carer in your working group. Including the perspectives and experiences of carers would strengthen this recommendation and provide a more comprehensive understanding of the support needed. It would be beneficial to integrate findings related to carer experiences or to highlight the lack of such perspectives in the current literature as a limitation that needs to be addressed. This will add depth to your recommendation and ensure that the voices of carers are adequately represented in your review. Which currently seem to be silenced.

2. You state, “Our research found that patients want to be actively involved in the decision-making regarding their treatment.” However, I did not see this mentioned elsewhere in the review. For consistency and thoroughness, it would be beneficial to ensure that this important finding is clearly presented and supported by data within the main body of your review. If this was indeed a finding from your research, please provide details on how this conclusion was reached and reference any relevant data or quotes from the studies. This will help to substantiate your statement and enhance the overall coherence of your review

Reviewer #2: General comments

The manuscript addresses a topical issue in mental health services which has implication for justice, equity and fairness within society as well as patient safety and satisfaction with care. The methodology adopted is innovative and the findings are likely to be of interest to the journal’s audience. The manuscript is generally well written and its flow is good. However, there are areas in which it can be further improved.

Introduction

1. This captures the essence of the background for the systematic review.

2. I suggest that the details of the 4 stages of the silences framework should be reserved for the methods section. In the background, these stages should only be raised as a conceptual and theoretical matter (with references) rather than a methodological description.

3. Please include the research question and aim/objectives of the review into the prose of the background rather than presenting them in bullets as they currently stand. This will make the background flow gently to its end rather than end abruptly as it has.

Methods

4. Please provide a reference for critical realism.

5. “We conducted a title and abstract search, and along with removing the papers from the additional exclusion criteria, we reduced the papers to 636”. Did the authors mean title and abstract screening?

6. “After the full paper review, 72 papers matched the streamlined inclusion criteria”. I thought this should read 32 papers and then when you add the later 2, you have 34. Authors should address this and amend as necessary.

7. I cannot find table 1 and it was difficult to assess the extent of data summarised on that table.

8. “The silences explored in this paper are just three out of many subthemes born from the workshops held throughout the process…”. On what basis were these three silences chosen as the key issues to be raised in this paper?

9. This raises the question of how the thematic analysis was approached in order to generate themes and sub-themes. It is important to understand how the authors randomly chose 50 papers from the second round of searching when in the end only 34 met inclusion for the systematic review? It would imply that some of the 50 papers did not make it into the systematic review but had been used to develop the themes used to interrogate the data for the systematic review as well as the silences attached to them.

10. Additionally, the thematic areas addressed in the narrative synthesis of the systematic review and the attached silences did not entirely follow the main themes identified from the table on themes shown in the supplementary material. It seems there were the final three themes – these should have been included in the table showing themes and sub-themes to make the data presentation clearer.

11. While I understand the need of the authors to contextualise their findings and show implication (and might fit more into the recommendations for reflexive thematic analysis), I would suggest that this being a systematic review primarily, the narrative synthesis of the data should be separated from “discussion” of some of the narrated findings so that there will be no repetitions as such. Currently, some aspects of the results would fit better into the discussion as they are not part of the narrative synthesis. In some cases (e.g. “culture, spirituality and religion”), the authors went as far as proffering recommendations for mental health services. This should be reserved for the discussion as it is “slowing down” the narrative in the results section.

12. I am not sure there is a need for two PRISMA flowcharts in the paper. The latter flowchart for the more recent search may be moved to supplementary material to avoid confusion. I will let the editor decide on this.

Discussion

13. The discussion should be amended by first discussing the findings from the systematic review and then the silences. This clarity will help the authors to situate their work in the existing literature as expected and also highlight the important gaps left by the silences and what they mean. Currently, some of the findings are not compared with previous literature and are mainly presented as advocacy points. The recommendations/implications raised will then make more meaning that way.

14. The limitations have not been rigorously discussed yet. While they have been raised descriptively, it is important to show how attempts were made to mitigate them. For example, why did the authors proceed to make recommendations from sources that did not feature in the systematic review? Was there a compelling reason to do so – e.g. they provided a good conceptual coverage, they were based on large datasets, sources are scare, etc.

References

15. Inconsistencies noted in the referencing style. Please ensure they conform to the journal style.

6. PLOS authors have the option to publish the peer review history of their article (what does this mean? ). If published, this will include your full peer review and any attached files.

**Do you want your identity to be public for this peer review?** For information about this choice, including consent withdrawal, please see our Privacy Policy .

Reviewer #1: No

Reviewer #2: **Yes: ** Adegboyega Ogunwale

---

## [Decision Letter · Decision Letter 1]

22 Nov 2024

PMEN-D-24-00122R1

A Co-Produced Review of the Experiences of Black male detention under mental health legislation: Challenging discrimination in Psychiatry using the Silences Framework.

PLOS Mental Health

Dear Dr. Heyes,

Thank you for submitting your manuscript to PLOS Mental Health. After careful consideration, we feel that it has merit but does not fully meet PLOS Mental Health’s publication criteria as it currently stands. Therefore, we invite you to submit a revised version of the manuscript that addresses the points raised during the review process.

The manuscript has been evaluated by two reviewers, and their comments are available below.

Reviewer 2 has some further requests for revision to the writing, interpretation of the results and the discussion. 

Could you please carefully revise the manuscript to address all comments raised?

We look forward to receiving your revised manuscript.

Kind regards,

Helen Howard

Staff Editor

PLOS Mental Health

Journal Requirements:

Additional Editor Comments (if provided):

Reviewers' comments:

Reviewer's Responses to Questions

**Comments to the Author**

1. If the authors have adequately addressed your comments raised in a previous round of review and you feel that this manuscript is now acceptable for publication, you may indicate that here to bypass the “Comments to the Author” section, enter your conflict of interest statement in the “Confidential to Editor” section, and submit your "Accept" recommendation.

Reviewer #1: All comments have been addressed

Reviewer #2: (No Response)

2. Does this manuscript meet PLOS Mental Health’s publication criteria ? Is the manuscript technically sound, and do the data support the conclusions? The manuscript must describe methodologically and ethically rigorous research with conclusions that are appropriately drawn based on the data presented.

Reviewer #1: Yes

Reviewer #2: Yes

3. Has the statistical analysis been performed appropriately and rigorously?

Reviewer #1: N/A

Reviewer #2: N/A

4. Have the authors made all data underlying the findings in their manuscript fully available (please refer to the Data Availability Statement at the start of the manuscript PDF file)?

Reviewer #1: Yes

Reviewer #2: Yes

5. Is the manuscript presented in an intelligible fashion and written in standard English?

Reviewer #1: Yes

Reviewer #2: Yes

6. Review Comments to the Author

Reviewer #1: Thank you for addressing al the concerns. Really enjoyed reading the manuscript again. It is such an honour to have read this important piece of research.

Reviewer #2: General comments

Thanks to the authors for their comprehensive responses to my comments. The paper reads much better now and necessary clarifications have been made for the prospective readers. Only a few minor issues remain.

1. Please recast “However, data shows that over double the amount of Black people with mental ill health die in custody than any other demographic…”to “…double the number or proportion of Black people…”

2. Please change “After duplications were removed…” to “After duplicates were removed…”. This is mainly a question of style or convention related to the process.

3. Please place MMAT in brackets.

4. Please recast “We held a total of four workshops in various guises.” to “We held a total of four workshops.” The use of the word “guises” could potentially confuse the reader into thinking that there was some covertness to the workshops or a level of concealment, on the part of the authors, regarding their real intentions for organising the workshops. This did not seem to be the case to me.

5. I would invite the authors to revisit the silences in ‘culture, spirituality and religion’ with regard to the following claim in the paper: “Those with lived experience on the project explained that spiritual practices and beliefs that exist outside of Judeo-Christian religions are often misunderstood and pathologized in mental health services; behaviour. Behaviour or beliefs that do not align with Western concepts of spirituality become a part of the individual’s mental health diagnosis or evidence of a mental illness entirely, which strips them of their autonomy by attributing a choice and a belief that could ground them to their mental state”. Were there instances in which Judeo-Christian beliefs themselves were contextually viewed as pathological by mental health services? This might provide a more nuanced or balanced understanding of the silences.

6. In the discussion segment, keep the references to the most recent or most influential in relation to the argument being proposed to avoid making the discussion look like the narrative synthesis all over again. For instance, the discussion around social capital has 16 references.

7. It would be helpful for the authors to situate their recommendations in the broader literature so that they could appeal to existing evidence base where possible/available.

8. Could the authors take a more critical approach to discussing the limitations? Lack of generalizability is key and should not be readily minimised. Limitation in relation to the quality of some of the sources e.g. grey literature that were (understandably) not subjected to quality appraisal should be briefly addressed. The predominance of the global North in the papers included (no papers/reviews from the global South really) ought to be discussed as this speaks to generalizability and perhaps global health inequalities between regions of the world (with the implication for sustainable development).

7. PLOS authors have the option to publish the peer review history of their article (what does this mean? ). If published, this will include your full peer review and any attached files.

**Do you want your identity to be public for this peer review?** For information about this choice, including consent withdrawal, please see our Privacy Policy .

Reviewer #1: No

Reviewer #2: **Yes: ** Adegboyega Ogunwale

---

## [Decision Letter · Decision Letter 2]

30 Jan 2025

A Co-Produced Review of the Experiences of Black male detention under mental health legislation: Challenging discrimination in Psychiatry using the Silences Framework.

PMEN-D-24-00122R2

Dear Dr Heyes,

We are pleased to inform you that your manuscript 'A Co-Produced Review of the Experiences of Black male detention under mental health legislation: Challenging discrimination in Psychiatry using the Silences Framework.' has been provisionally accepted for publication in PLOS Mental Health. Please accept my sincere apologies for the delays experienced during the handling of this paper. This is not reflective of the usual process at PLOS Mental Health and I will do my best to ensure there are no repeats of this.

Best regards,

Karli Montague-Cardoso

Executive Editor

PLOS Mental Health

Reviewer Comments (if any, and for reference):

Reviewer's Responses to Questions

**Comments to the Author**

1. If the authors have adequately addressed your comments raised in a previous round of review and you feel that this manuscript is now acceptable for publication, you may indicate that here to bypass the “Comments to the Author” section, enter your conflict of interest statement in the “Confidential to Editor” section, and submit your "Accept" recommendation.

Reviewer #2: All comments have been addressed

2. Does this manuscript meet PLOS Mental Health’s publication criteria ? Is the manuscript technically sound, and do the data support the conclusions? The manuscript must describe methodologically and ethically rigorous research with conclusions that are appropriately drawn based on the data presented.

Reviewer #2: Yes

3. Has the statistical analysis been performed appropriately and rigorously?

Reviewer #2: N/A

4. Have the authors made all data underlying the findings in their manuscript fully available (please refer to the Data Availability Statement at the start of the manuscript PDF file)?

Reviewer #2: Yes

5. Is the manuscript presented in an intelligible fashion and written in standard English?

Reviewer #2: Yes

6. Review Comments to the Author

Reviewer #2: Thanks for your diligence in addressing all previous concerns raised. This is a nice piece of work which contributes to the literature on the mental health detention experiences of black males and provides actionable recommendations for improving the experience as well as combatting discrimination more broadly in mental health services.

7. PLOS authors have the option to publish the peer review history of their article (what does this mean? ). If published, this will include your full peer review and any attached files.

**Do you want your identity to be public for this peer review?** For information about this choice, including consent withdrawal, please see our Privacy Policy .

Reviewer #2: **Yes: ** Adegboyega Ogunwale
